# Stability and Safety Traits of Novel Cyclosporine A and Tacrolimus Ophthalmic Galenic Formulations Involved in Vernal Keratoconjunctivitis Treatment by a High-Resolution Mass Spectrometry Approach

**DOI:** 10.3390/pharmaceutics12040378

**Published:** 2020-04-20

**Authors:** Daniele Giovanni Ghiglioni, Piera Anna Martino, Gaia Bruschi, Davide Vitali, Silvia Osnaghi, Maria Grazia Corti, Giangiacomo Beretta

**Affiliations:** 1Fondazione IRCCS Ca’ Granda Ospedale Maggiore Policlinico di Milano, Via Francesco Sforza, 28, 20122 Milan (MI), Italy; daniele.ghiglioni@policlinico.mi.it (D.G.G.); silvia.osnaghi@policlinico.mi.it (S.O.); mariagrazia.corti@policlinico.mi.it (M.G.C.); 2Department of Veterinary Medicine, Università Degli Studi di Milano, Via Celoria 10, 20133 Milan (MI), Italy; piera.martino@unimi.it; 3Department of Clinical and Community Sciences, University of Milan, Via della Commenda, 19, 20122 Milan (MI), Italy; 4Bruttomesso Pharmacy, Galenic Laboratory, Piazza Guglielmo Marconi, 20, 26013 Crema (CR), Italy; lab@farmaciabruttomesso.it; 5Department of Environmental Science and Policy, Università degli Studi di Milano, Via Mangiagalli 25, 20133 Milan (MI), Italy; giangiacomo.beretta@unimi.it

**Keywords:** cyclosporine A, tacrolimus, ophthalmic formulations, vernal keratoconjunctivitis

## Abstract

In this study, a sensitive quantitative method based on high performance liquid chromatography combined with high-resolution mass spectrometry, Q Exactive^TM^-Orbitrap^®^ was set up and applied for the determination of the immunosuppressor agents cyclosporine A and tacrolimus in novel ethanol-free ophthalmic formulations for the treatment of Vernal keratoconjunctivitis. Different storage parameters in terms of storage temperatures and practical usage conditions were investigated to assess the stability of all formulations during shelf life simulating the real conditions as well to confirm the feasibility of use of ethanol-free products. The methodology was linear (r^2^ = 0.995) over the concentration range 0–200 ng/mL, and its selectivity, precision, accuracy and recovery were all within the required limits. Under different conditions (storage period 0–90 days, 5–25 °C, unopened/usage simulated conditions), our results revealed that both active pharmaceutical ingredients (API) show satisfactory stability up to 30 days of storage/usage, with a significant and consistent concentration decline of cyclosporine A after this time point when its hydroalcoholic formulation was kept at 25 °C.

## 1. Introduction

Cyclosporine A and tacrolimus are immunosuppressor agents with therapeutic indications for the treatment of several pathological conditions such as ocular Behçet’s syndrome, endogenous uveitis, psoriasis, atopic dermatitis, rheumatoid arthritis, active Crohn’s disease and nephrotic syndrome, as well as prophylaxis and treatment of transplant rejection. Among these pathologies, Vernal (from the Latin word for “spring”) keratoconjunctivitis (VKC) is a bilateral, asymmetric, chronic anterior surface disease, classified among chronic allergic eye diseases predominantly of the pediatric age, with still unknown immunopathogenesis [1,2]. Its symptoms are similar but significantly amplified comparing to those of other ocular allergic forms [3]. The subjective symptomatology is characterized, with variable intensity, by itching, the development of a thick and filamentous mucoid secretion, photophobia, burning, lacrimation, foreign body sensation, “redness” (hyperemia, often complained of as a “subjective symptom”), palpebral pseudo-ptosis and also sometimes pain, which can seriously affect the quality of life of affected children. The current VKC therapy involves as first line option the application of the same topical agents used in the treatment of other forms of allergic conjunctivitis. However, the severity classifications of VKC present in the literature are not always unambiguous, conditioning therapeutic schemes that differ from center to center [4,5,6,7,8,9,10,11]. Topical antihistaminic agents, mast cell stabilizers and dual action eye drops relieve itching and inhibit the release of mast cell mediators but are effective only in cases of mild VKC [12] or have given conflicting results in patients from different geographical areas [13,14]. Topical nonsteroidal anti-inflammatory drugs (NSAIDs) also reduce ocular inflammatory signs, decreasing the use of corticosteroids, without being able to replace them; sometimes, they appear to be associated to corneal damage [13]. Topical corticosteroids are the most effective treatment for moderate to severe VKC forms. However, their long-term use must be strictly limited and carefully monitored due to their potential complications (glaucoma, cataract, ocular hypertension, secondary bacterial or viral eye infections such as herpetic keratitis) [15]. In this context, to avoid the prolonged use of cortisone drugs, galenic ophthalmic preparations containing cyclosporine A (0.5–1% up to 2%) and tacrolimus (0.1%)-based eye drops have been developed and studied in various double-blind placebo trials, proving to be effective in various concentrations for the treatment of moderate and severe VKC [16,17]. Regarding high-dose cyclosporine preparations (10–20 mg/mL), they are prepared usually as oil ointment or artificial eye drop solution-based formulations starting from injectable cyclosporine A preparation containing ethanol as co-solvent. The presence of ethanol leads to patients’ discomfort at product instillation, especially critical in pediatric patients. In addition, commercial ophthalmic products containing cyclosporine A at lower concentration (e.g., 0.5 mg/mL, Restasis^®^), are of dubious efficacy in the treatment of VKC [18] and they are highly expensive (e.g., cyclosporine 1.0 mg/mL, Ikervis^®^). In some cases, these preparations are not registered for the treatment of VKC or they are recognized as VKC orphan drug (e.g., cyclosporine 1.0 mg/mL, Verkazia^®^) [19,20], but still not commercially available or highly expensive. Some researchers have already investigated some quality traits of tacrolimus and cyclosporine-based eye drop formulations during shelf life. These studies are limited to the elucidation of the active compound stability in unopened products or to description of their degradation products [21]. Considering the above mentioned considerations, the aim of the present study was to evaluate the reliability of novel ethanol-free cyclosporine A and tacrolimus galenic eye drop formulations through the assessment of their stability and microbiological safety, in comparison to conventional hydroalcoholic 1% cyclosporine A and 0.1% tacrolimus formulations prepared by dilution of commercial Sandimmun^®^ and Prograf^®^, respectively. Stability was monitored under both shelf life and simulated usage conditions, using a high-performance liquid chromatography coupled to high-resolution mass spectrometry (Q Exactive^TM^ Orbitrap^®^). The feasibility of the use of ethanol-free cyclosporine A and tacrolimus galenic eye drop galenic preparations represent a crucial issue in order to minimize the side effects during the administration in children populations thus facilitating the compliance of therapy especially when chronic treatment is required.

## 2. Materials and Methods 

### 2.1. Chemicals and Reagents

Methanol, ammonium formate and the internal standard Proadifen Hydrochloride (SKF-525A), were purchased from Merck KGaA (Darmstadt, Germany). Ultra-pure water was produced using a Milli-Q system (Millipore, Merck KGaA, Darmstadt, Germany). Tacrolimus (FarmaQuimica Sur, Malaga, Spain), Cyclosporine A (Acef, Fiorenzuola d’Arda, Italy), Sandimmun (Novartis, Basilea, Switzerland), Prograf (Astellas Pharma Inc., Tokyo, Japan) Lacrimart (Baif International, Genova, Italy), polyvinylpyrrolidone (PVP; Acef, Fiorenzuola d’Arda, Italy), injectable-grade water (Fresenius Kabi, Verona, Italy), and Cremophor RH 40 (Acef, Fiorenzuola d’Arda, Italy).

### 2.2. Standard Solutions

The working solutions of tacrolimus, cyclosporine A and of the internal standard for method validation and calibration curves construction were prepared daily in methanol from corresponding stock solutions (1 mg/mL) and stored at −20 °C until use.

### 2.3. Cyclosporine A and Tacrolimus Ophthalmic Formulations

Different galenic formulations were investigated in this study to assess the feasibility to minimize ethanol or lipid source used in order to prevent the children’s patient side effects at instillation phase with better contort as final result. Several hospital pharmacies are involved in ophthalmic preparations, based on the classical approach in which the active compounds are carried by using injectable solutions as well as by using new promising protocols in which the eye drops are prepared as micellar by using lipid vehicle, such as polyols castor oil as an example.

The present study design included two ophthalmic formulations based on two active substances: high dose cyclosporine A preparations and tacrolimus comparing two different preparation protocols.

#### 2.3.1. Cyclosporine A-based Ophthalmic Formulations (1%)

Classical formulation (SAND): Sandimmun^®^ (concentrated injectable solution, 50 mg/mL) was diluted 2:8 vol/vol in commercial artificial drops Lacrimart^®^ and homogenized under agitation to obtain a clear lipid solution as the most common galenic protocol.Novel ethanol-free formulation (CSA): cyclosporine A 0.1 g, polyvinylpyrrolidone 0.2 g Cremophor RH-40 1 g, 10 mL injectable-grade water were used as ingredients.

#### 2.3.2. Tacrolimus-based Ophthalmic Formulations (0.1%)

Classical formulation (PROG): Prograf^®^ (5 mg/mL) was diluted 2:8 vol/vol in in commercial artificial drops Lacrimart^®^ and homogenized under agitation to obtain a clear limpid solution as the most common galenic protocol.Novel ethanol-free formulation (TAC): Tacrolimus 0.01 g, polyvinylpyrrolidone 0.2 g Cremophor RH-40 1 g, 10 mL injectable-grade water were used as ingredients.

All formulation details in terms of their composition are summarized in Table 1. The rationale adopted for novel eye drop formulation is based on the difficulty of solubilization of organic macromolecules such as CSA and Tacrolimus. In addition, the classical formulations involved simply the dilution of injectable ampoules designed for other routes of administration and with an excipient suitable for those routes (intramuscular and others). The resulting eye drops contain 100 mg of ethanol in 10 mL of final eye drops, which can lead to a burning side effect. The first formulation step is based on the identification of an emulsifier suitable for the ocular pathway supported by an artificial tear with chemical characteristics that can be considered a “co-emulsifier”. In fact, polyvinyl alcohol and Na Jaluronate are not suitable if compared to PVP, which has at its base functional ketone groups, which is also present on the whole structure of the two active ingredients investigated.

For CSA and TAC formulation, all the constituents of the eye drops are weighed in an aseptic chamber inside a nonsterile container Turbo emulsifier (SAMIX ES 500, Farmalabor Tech, Assago, Milan, Italy). Once weighed, they are mixed for 5 min at 2340 rpm, the obtained emulsion is placed under a laminar flow hood with a germicidal UV lamp for 30 min. Once the flow conditions are activated and the aseptic environment is rendered, a 10 mL aliquot of the emulsion is taken, and filtered by using sterile syringe equipped with a 0.22 micron Millipore filter for aqueous solutions filtration, obtaining sterile eye drops. The final ophthalmic products obtained were clear lipid solutions. All final ophthalmic solutions were packaged in a standard, white, opaque low-density polyethylene (LPDE) squeezable 10-mL bottles closed with cap (ACEF, Fiorenzuola d’Arda, Italy). Containers commonly used for ophthalmic products include glass containers and polyethylene containers. Glass containers and polyethylene (PE) containers are recognized to be superior in maintaining stability of ophthalmic preparations. LDPE containers were used to store all involved formulations in the present research. The container quality represents a crucial point since the purity of a medicinal preparation may also change during storage due to leaching of chemical or chemicals into the drug preparation from the container materials, from the labels on the containers, or from the environment where the packaged ophthalmic product is stored [22]. Thus, containers used for packaging medicinal preparations can significantly affect the stability and purity of the preparations as well. All LPDE containers used were certified for storage of ophthalmic drug regarding criteria of water loss, environmental stress cracking resistance (ESCR). The LPDE material are certified according European Pharmacopoeia (“Polyolefines and Polyethylene without additives for containers for preparations for parenteral use and for ophthalmic preparations” (8th edition – 2014). They were tested by the producer in accelerated conditions of temperature and relative humidity (40 °C ± 2 °C/ NMT 25% R H for plastic) during a three-month estimation drug assay. Plastic materials were tested according the migration test to assess the safety conditions toward compounds that can be transfer from plastic material to eyedrop. The stability studies were finally conducted on the drug substances packaged in a container closure system that is the same of those proposed for storage and distribution.

In order to confirm the stability of all formulations involved in the present research (Tacrolimus- and CSA-based), they underwent High Resolution Mass Spectrometry (LC-HRMS) evaluation to detect their concentration of active compounds at time t0 (10 mg/mL for CSA and 1 mg/mL for Tacrolimus eyedrops). For each formulation, concentration reaching more than 99.8% of the active compound compared with the declared one was considered as satisfactory criteria. In its final form, the ethanol-free formulations consisted of an isotonic aqueous solution with micelles of Cremophor® solubilizing the cyclosporine and Tacrolimus with the surfactant and polymer aiding solubility.

During all experiments, all of the multidose eyedroppers were emptied into polycarbonate test tubes and the solutions were visually inspected under white light in front of a matte-black panel and a non-glare white panel. Aspect and colour of the solutions were noted, and a screening for visible particles, haziness, or gas development was performed.

### 2.4. Stability of Cyclosporine A and Tacrolimus Ophthalmic Formulations

Different storage conditions were tested to investigate their stability, as the literature is scarce:-Real use of simulated conditions in opened products: all formulations underwent analysis, testing two different temperatures (25 °C as room temperature and 5 °C ± 2 °C as refrigerated) during 90 days of storage in which all bottles were regularly gently shaken and opened (4 s each time, three time per day) in order to simulate the real use conditions. As partially investigated, the role of temperature in maintaining active substances stability can play a central role for quality of preparations.-Shelf-life investigation on unopened products: all formulations underwent active compound analysis during the 90-day testing. The unopened bottles were assessed to verify the decay of cyclosporine A and tacrolimus to assess the possibility to prepare in advance the formulations leading to several advantages for galenic pharmacies in term of stock feasibility.-Stressed temperature conditions during delivery: all formulations underwent active compounds analysis, simulating an unfavorable temperature condition during transport (simulating courier time delivery), since most of the therapies are also administrated during summer period. 40 °C was selected as the stress parameter compared with the refrigerated one during 4 days by using unopened bottles.

All bottles containing eyedroppers were stored lying down horizontally at controlled refrigerated temperatures of (Whirlpool refrigerator) at 5 °C ± 2 °C (temperature measured with a Testo 175-T1 probe; Testo SARL, Forbach, France) or in a climate chamber (BINDER GmbH, Tuttlingen, Germany) at 25 °C ± 2 °C and 60% residual humidity (RH), until analysis.

### 2.5. Sample Processing

All samples were diluted in ammonium formate 20 mM: methanol (70:30) to obtain a theoretical final concentration of 100 ng/mL (1:10000 for tacrolimus and 1:100000 for cyclosporine preparations respectively). Internal standard SKF-525A was added at the same concentration in the final volume of 1 mL, inserted in 2 mL glass autosampler vials and submitted to HPLC-HRMS analysis.

### 2.6. HPLC-HRMS Analysis

Separations were done using an HPLC system (Thermo Fisher Scientific, San Jose, CA, USA), equipped with a Surveyor MS quaternary pump and a degasser, a Surveyor AS autosampler with column oven, and a Rheodyne valve with a 20-μL loop. A Synergi Hydro-RP reverse-phase HPLC column (150 × 2.0 mm, i.d. 4 μm), with a C18 guard column (4 × 3.0 mm; Phenomenex, Torrance, CA, USA) were used. The mobile phase consisted of a binary mixture of solvents A (20 mM aqueous ammonium formate) and B (MeOH). The elution started with 30% B, which increased to 95% in 5 min and kept in these conditions until 10 min. The initial conditions were reached in the 11th min, with an equilibration time of 7 min. The run was performed at 0.3 mL/min.

The detector was a Thermo Q-Exactive Orbitrap™ (Thermo Scientific, San Jose, CA, USA), equipped with a heated electrospray ionisation (HESI) source. Capillary and vaporiser temperatures were set at 330 °C and 280 °C, respectively, while the electrospray voltage was set at 3.50 kV, operating in positive mode. The sheath and auxiliary gas were set at 35 and 15 arbitrary units (AU). Xcalibur 3.0 software (Thermo Fisher Scientific, San Jose, CA, USA) was used to control the HPLC-HRMS system. The full scan (FS) acquisition was combined with a data-independent acquisition (DIA) strategy, providing the MS2 spectra for a confirmatory response, based on the inclusion list. The FS resolution was 70,000 FWHM, the selected scan range was 200–1250 m/z; the automatic gain control (AGC) was set at 1 ×106, and the maximum injection time was 300 ms. The DIA segment operated in positive mode at 35,000 FWHM. The AGC target was set to 2 ×104, with the maximum injection time of 100 ms. The isolation window was of 1 m/z. Fragmentation of the precursors was obtained with two-step normalized collision energy (10 and 15eV). Detection of the analytes was based on the retention time (RT) of the target compounds, the calculated exact mass of the protonated molecular ions, and at least one specific and typical fragment. Acquisition data were recorded and elaborated using Xcalibur™ software (Thermo Fisher). All determinations are conducted in triplicate.

### 2.7. Method Validation

Validation was performed according to European Compliance Academy (ECA) and European Medicines Agency (EMA) guidelines [23] and in agreement with international guidelines for analytical techniques for the quality control of pharmaceuticals [24]. Specificity and selectivity, linearity, limit of quantification (LOQ), precision and accuracy were evaluated.

### 2.8. Microbiological Assay

Forty-eight ophthalmic solutions were cultured for bacteria; in particular, the solutions were tested at the opening and after 30 days using regularly opened bottles; moreover, the packs after opening were stored at 4 °C and at room temperature and were tested at both conditions. For each solution, 20 drops were evaluated (500 µL) and placed in 4.5 mL of Brain Heart Infusion Broth (Microbiol, Cagliari, Italy) and incubated at 35 ± 2 °C for 48 h under aerobic conditions. Each test was performed in duplicate. After the incubation time, 100 µL of the incubated broth for each sample was streaked onto blood-agar plates (Microbiol, Cagliari, Italy) and incubated as described above.

### 2.9. Data Analysis

Stability for all investigated formulations was defined as the time at which 90% of the initial concentration of Tacrolimus and Cyclosporine A active compounds remained (t90); the initial concentration (time point 0) was considered to be 100%. Active compounds concentration was finally expressed as the percentage of the initial active compound concentration remaining at each sampling time. Statistical evaluation of the data was conducted by using SPSS software, Version 24; Chicago, IL: SPSS Inc; 2002. Analysis of variance (ANOVA) was applied to the data and significant differences were determined by one-way ANOVA and Student–Newman–Keuls (SNK) as a post-hoc test to evaluate differences among different formulations and storage times. An effect was considered significant at the 5% level (p < 0.05).

## 3. Results

### 3.1. Tacrolimus and Cyclosporine A Quantification in Different Galenic Preparations

The formula of the investigated active compounds cyclosporine and tacrolimus, with the exact theoretical mass of the parents and the diagnostic transition, used to confirm them were reported in Table 2. The extracted parent ion chromatograms, acquired from Full Scan (FS) analysis, the peak of the main fragment and the mass spectra are also shown in Figure 1A. Bearing in mind that there is limited information regarding the structure of main fragments obtained by high-resolution mass spectrometry, we have proposed the fragmentation pattern for each of compounds enrolled in this study. The mild two step collision energy (10 and 15 eV) turns to be fundamental for reproducible and characteristic bond cleavage and molecular rearrangements (Figure 1B).

The chromatographic method used is linear for concentrations ranging from 12 to 28 μg/mL. Concerning the validation results, the mean linear regression equation obtained was *y* = 67.196x + 10.069 where *x* is the cyclosporine A concentration and *y* the surface area of the corresponding peak. The corresponding correlation coefficient was 0.997. The accuracy, expressed as percentage of mean calculated to nominal concentration ranged from 93% to 105% for tacrolimus and from 90% to 107% for cyclosporine A. The inter-day precision was 8% and 9% for cyclosporine and tacrolimus, respectively.

### 3.2. Stability of Tacrolimus and Cyclosporine A in Eye Drop Formulations under Usage-simulated Conditions

The 90-day stability trend for all formulations was presented in order to evaluate the active substances decay, with particular attention to cyclosporine A (CSA) and tacrolimus (TAC) ethanol-free eye drops. All observations were conducted in simulated usage conditions in which the samples were opened at time 0 and monitored during prolonged storage of 90 days as described in material and method section. The relative stability of tacrolimus in both Prograf^®^ and TAC formulations was evaluated under simulated usage conditions, at the storage temperatures of 25 °C (room temperature) and 5 °C (refrigeration).

In all cases, at the end of storage period, the concentration decay was within the 10–20% range (Figure 2A), in accordance with the results reported by Ezquer-Garin et al. [25], that described a suitable stability of 0.3% tacrolimus in an eye drop formulation stored at 5 °C, but with a significant decline in its concentration (<90%) after 28 days of storage at 25 °C (*p* < 0.05) for all formulations involved in the present study. By contrast, a higher significant degradation rate was observed considering all storage times for cyclosporine A in the Sandimmun formulation already at the first time point tested (40% degradation, *p* < 0.05), reaching 80% after 30 days of storage at room temperature (Figure 2B, *p* < 0.01).

Conversely, the degradation time-course of cyclosporine A in Sandimmun stored at 5 °C was similar to that of cyclosporine A in CSA formulation, regardless of the storage temperature without significant differences among different eye drops.

These results were in good accordance with those reported by Chennel et al. [22], that evaluated the stability of cyclosporine A in an eye drop formulation even if limited at 30 days under usage conditions, observing no significant degradative effect. However, in this last study the evaluation of active substances behavior was not extended to prolonged storage period and to simulated usage conditions as done in the present study, that is important since the frequent instillations required by the therapy regimen, often involve the bottle opening and closing more than 3 time per day that can lead to cyclosporine A and tacrolimus degradation trend more consistent especially after 30 days. Regarding visual inspection, all samples stayed limpid and with a slight yellow tinge throughout the study, for the tested formulations and conservation temperatures, and there was no appearance of any visible particulate matter, haziness or gas development.

### 3.3. Shelf-life of Tacrolimus and Cyclosporine A Formulations in Unopened Products

The 60 days stability of unopened formulations kept at refrigerated storage conditions was determined to confirm the suitability of eye drops to be stored for an extend time period. The temperature 5 °C was selected as it represents the best storage condition to ensure both cyclosporine A and tacrolimus stability. The shelf-life of all four formulations was similar when vials were first opened. The tacrolimus and cyclosporine A concentration variation was <20% in respect to the T0 concentration (100%) even at the longest storage time period tested of 60 days (Figure 3). The highest and significant degradation rate was observed in CSA considering its storage times, if compared with other formulations (*p* < 0.05), but remaining below 12% active compound loss. This could represent a great advantage especially for hospital galenic pharmacies in terms of the possibility to set up product stocks, saving working time to guarantee products timely availability.

The active compounds decay in all formulation was also investigated after 30 and 60 days of unopened storage to assess a variation of Cyclosporine A and tacrolimus testing the real use conditions over 30 days by opening and shacking the solutions every day. All bottles once opened were subjected to two storage conditions, at room and refrigerated temperature, monitoring the decay at 0, 7, 15 and 30 days shelf-life for formulation after 30 days unopened and at 0, 15 and 30 days for formulations after 60 days unopened. This is crucial to assess and confirm the feasibility to set up stock formulations saving consistent losses of active compounds. All results are reported in Figure 4 and Figure 5.

Considering all storage times tested, significant losses of cyclosporine A in Sandimmun^®^ (SAND) and cyclosporine A (CSA) samples were measured in vials opened both after 30 days and after 60 day of *unopened* storage (*p* < 0.01), in formulations stored at room temperature (25 °C, −60% as maximum decay trend) confirming the role of temperature to preserve active compounds losses.

Concerning Tacrolimus formulations is was been observed how all storage conditions had minor influence even if refrigerated storage were able to guarantee the minimum decay trend both after 30 and 60 days of unopened conservation at 4 °C there was a loss of activity of 15–20%, respectively), confirmed by a nonsignificant variations. As final consideration, the present experiment confirms the possibility to set up stock of cyclosporine A and Tacrolimus eyes drops, considering their use after 30 days of unopened storage assuring acceptable active compounds losses less than 20% compared to their initial concentration.

### 3.4. Effect of Delivery-simulated Conditions on Tacrolimus and Cyclosporine A

With the aim to establish any detrimental effect of extreme storage conditions that can take affect APIs concentration under non-refrigerated expedition of the medicinal products to the patient, their stability after 24 h and 72 h time-points at 40 °C was monitored. This is important since the delivery of products may be done in summer periods by courier transportation without refrigeration delivery as well. The overall trend observed for tacrolimus and cyclosporine A preparations, respectively, is presented in Figure 6.

The results confirm the negative impact of high temperatures also during the 72 h of simulated delivery, with significant decay (*p* < 0.01) observed after 48 h only for products stored at 40 °C, confirming the necessity of refrigerated storage conditions also during transportation as best practice to keep the active compounds of formulations stable as well.

### 3.5. Microbiological Stability

All samples showed no bacterial charge (sterile), for all formulations tested during 30 storage days. The results showed that no bacterial contamination was evident in all opened ophthalmic solutions maintained both at 4 °C and at environmental temperature, also assessing the feasibility of the use of new ethanol-free formulations as well. The microbiological safety in terms of sterility represents a critical element since it represents a mandatory requirement for possible application as ocular preparation.

As previously reported in literature, the risk of eye drops contamination is related to reuse of these therapeutic presidium and to its duration of use [26,27,28]. As such, it should be pointed out that further investigations would be needed to evaluate whether multi-dose eye medication could be contaminated by bacteria attributable to a potential incorrect user handling.

## 4. Discussion

In the present study, the stability of the immunosuppressor agents cyclosporine A and tacrolimus in ophthalmic formulations with and without ethanol have been evaluated using a highly selective and sensitive methodology based on high-resolution mass spectrometry.

Previous recent studies on the stability of cyclosporine A (10–20 mg/mL) in an ethanol-depleted micellar formulation stored at 25 °C and 5 °C for one month with monitoring by High Performance Liquid Chromatography (HPLC-UV), found a satisfactory stability with no significant difference in the concentration of this API comparing to its starting concentration at the end of the storage period [23]. However, the authors reported that temperatures above 25 °C may induce potential instability at higher temperatures.

In another recent study, no significant decrease of tacrolimus in eye drop formulation (0.3%) was found after 85 days of storage under refrigerated conditions (5–25 °C). Conversely, when tacrolimus solution was stored at 25 °C, its concentration decreased below 90% of the starting concentration already after 28 days [25]. At present, scarce information on the stability of these pharmaceutical drugs in ethanol-free formulations, as investigated in the present work, is available. The results of the major efforts for the definition and confirmation of cyclosporine A and tacrolimus as effective drugs for VKC treatment (efficacy and side effects) are available from clinical studies and trials as summarized in Appendix A.

VKC is widespread in areas with a warm and temperate climate such as the Mediterranean basin, north and west of Africa, the Middle East, the Anatolian peninsula, the Arabian Peninsula, parts of India, Pakistan, Japan, and Central and South America.

Indicated among the rare diseases in the “Orphanet Journals” of January 2019 with a prevalence of 32/100000, VKC could actually have a greater prevalence also in Europe, as hypothesized for a long time; in East Africa, the VKC affects more than 5% of children of school age [29], and in some populations of Africa (Ethiopia) a prevalence of up to 5.2–7.3% has recently been estimated. With Orphanet’s European estimates in the northern Italian region of Lombardy, there would be over 3000 people affected by VKC, almost 20,000 in Italy and 240,000 in Europe. VKC mainly affects individuals of preschool age (3–5 years) up to the end of the second decade of life, with a peak incidence between 11 and 20 years [30]. It mainly affects the male sex, with a male–female ratio ranging from 3 to 1 to 7 to 1. A family history of atopy is present in 40% of subjects [31].

Chronic conjunctival inflammation and the release of inflammatory and keratolytic substances determine an increased fragility and sensitivity of surface ocular tissues, predicting corneal outcomes, such as superficial, localized or diffuse punctate keratitis, which over time can turn into an ulcer “a shield” (mainly produced by a toxic immune-allergic mechanism) or in corneal abrasions and in superficial proliferation of newly formed vessels [32]. “Shield” ulcers affect about 10% of patients, appear especially in periods of greater intensity of symptoms and are considered to be feared complications [32]. In fact, they may result in elevated astigmatism, keratoconus or more rarely in corneal perforation. There is a variable percentage of patients from 8% to 15% that is substantially resistant to treatment with cyclosporine A, at least for the central 3–4 months of the spring–summer period [33]. In the most serious cases, resistant to cyclosporine A, studies on the efficacy and safety of a local treatment with tacrolimus are in progress, prepared in a similar way to those used for cyclosporine A, which so far have given favorable results, so much so that tacrolimus has been declared, like cyclosporine A, an orphan drug for VKC by the EMEA [23,33].

The risks inherent to the use of galenic preparations are known and linked to the absence of a standardization in the preparation, to which is also added the before mentioned symptomatology.

In recent years, cyclosporin dissolved in polyoxyl 40 stearate (MYS-40) has been registered for the treatment of allergic conjunctivitis not responsive to common therapies is on the market in Japan: it uses a 0.1% cyclosporine A concentration [34]. Due to its aqueous composition, the distribution of the drug would seem to be about 3 times better than the emulsion in oil. This 0.1% formulation has been tested in Japanese children with VKC and atopic keratoconjunctivitis AKC, and patients with VKC have been shown to respond within one month of starting the administration and, after 6 months of therapy, they have only minimal symptoms. Cyclosporine-based drugs are available on the market at lower concentrations (0.05%) that have been found to have contrasting efficacy in VKC control [8,35]. In this context, the results recently reported by Leonardi et al. (2019) [19] and by Bremond Gignac et al. (2019) [20] would confirm the efficacy of 0.1% cyclosporin A, due to the solution in which they are emulsified for severe VKC treatment both in children and adolescents.

In the present research, we tested galenic preparations at a higher concentration that has proven to be certainly effective so far (1%) [6]. Then it would be desirable to find effective preparations characterized by a better solubility and distribution on the ocular surface level, in order to reduce possible side effects. It also crucial to develop galenic preparations that do not cause intense local burning (such as the galenics used so far) with lower costs than the eye drops currently available on market to obtain patient contort and cost saving as main results.

## 5. Conclusions

Our study shows how the preparation of cyclosporine A powder in galenic, ethanol-free, eye drop (ingredients) at the concentration of 1% w/vol (CSA) and tacrolimus in ethanol-free, galenic eyedrop (0.1% w/vol, TAC) show satisfactory stability profiles; are stable, sterile and potentially as effective as traditional formulations, and also ensure good tolerability by the patient by not containing the alcohol commonly used to ensure the stability of the compounds. In addition, we also underline how the galenic preparations are less expensive if compared with available commercial drugs, resulting in pharmacies saving costs, an especially important issue when they are inserted in a hospital context. However, it is necessary to pass to the clinical trial on patients to assess both the real efficacy of the preparations as well as the side effect reductions reported by a clinical trial.

## Figures and Tables

**Figure 1 pharmaceutics-12-00378-f001:**
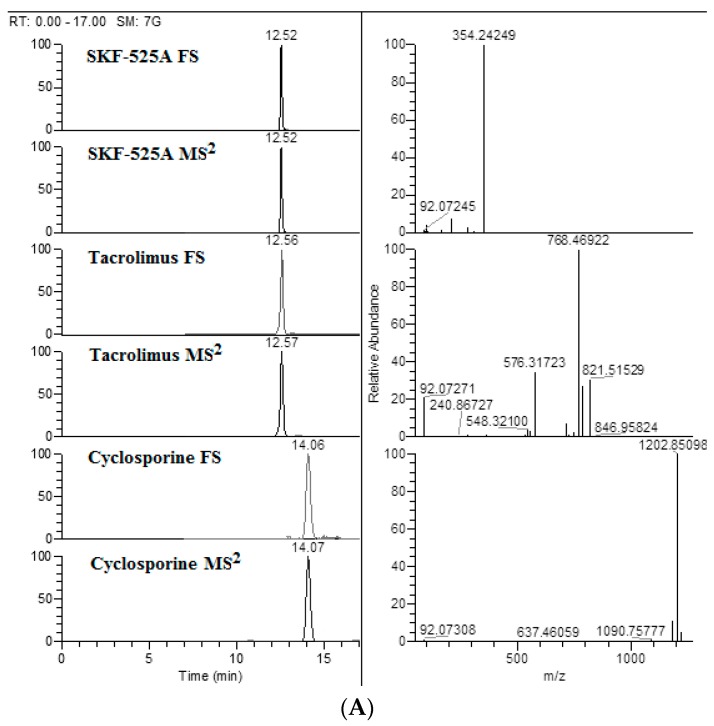
(**A**) Ion chromatograms extracted from full scan (FS), chromatograms of main fragment obtained by mass spectra MS^2^ acquisition and mass spectra of proadifen (SKF-525A), tacrolimus and cyclosporine A (100 ng/mL). (**B**) q-Exactive Orbitrap high-resolution mass spectrum and proposed fragmentation pathway of SKF-525A, tacrolimus and cyclosporine A, acquired in Data Independent Acquisition (DIA) mode with a two-step normalized collision energy (10 and 15 eV).

**Figure 2 pharmaceutics-12-00378-f002:**
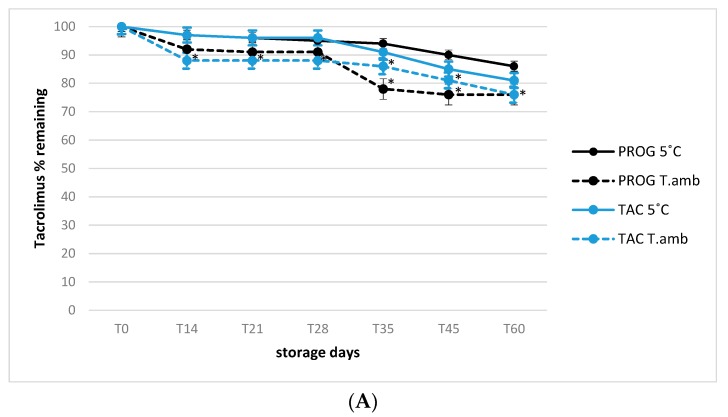
(**A**) Active principle stability in 0.1% tacrolimus (A) eye drops formulations stored at 5 °C and 25 °C (T.amb) under user simulated conditions (see Experimental for details); * *p* < 0.05, T.amb versus 5 °C storage temperature. (**B**) Active principle stability in 1.0% cyclosporine A (B) eye drops formulations stored at 5 °C and 25 °C (T.amb) under user simulated conditions (see Experimental for details); * *p* < 0.05, T.amb versus 5 °C storage temperature.

**Figure 3 pharmaceutics-12-00378-f003:**
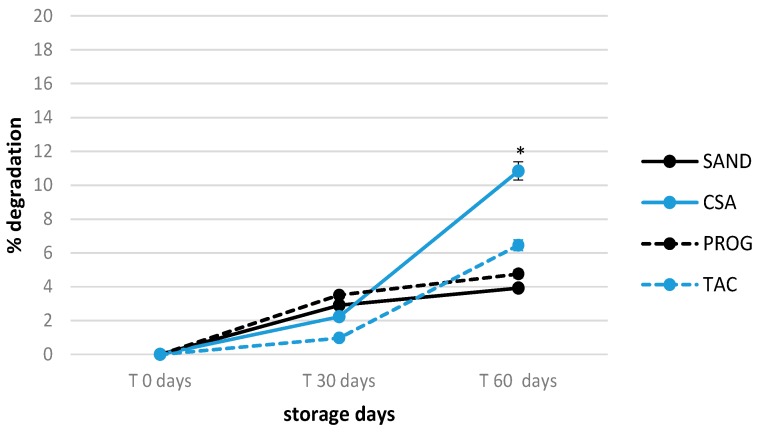
Active principle stability during 60 days in 0.1% tacrolimus and 1.0% cyclosporine A unopened eye drop formulations stored at 5 °C; * *p* < 0.05, T.amb versus 5 °C storage temperature.

**Figure 4 pharmaceutics-12-00378-f004:**
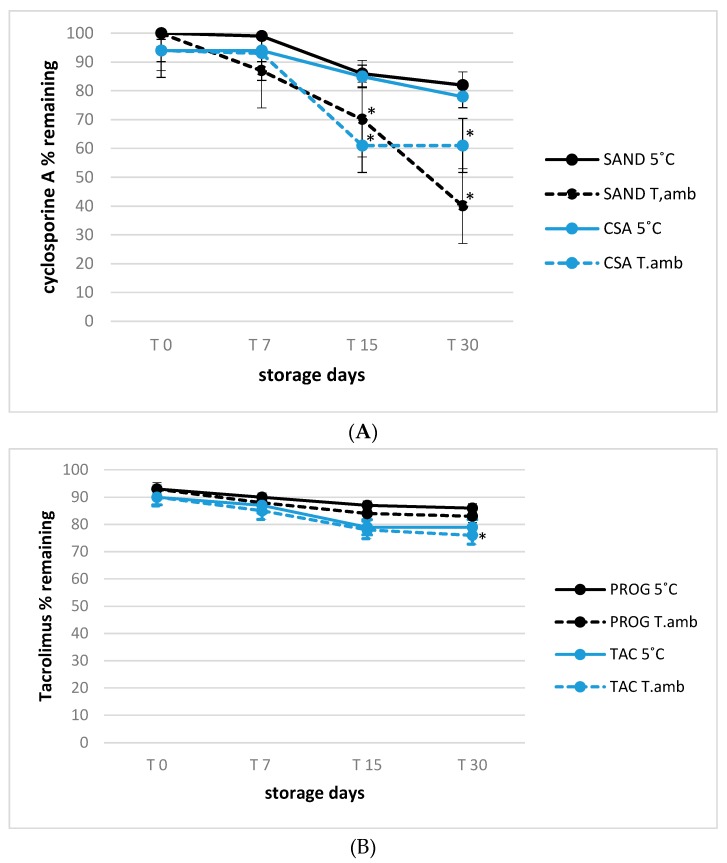
(**A**) 1.0% cyclosporine A formulations stability for 30 days shelf-life after 30 days unopened storage; * *p* < 0.05, T.amb versus 5 °C storage temperature. (**B**) 0.1% tacrolimus formulations stability for 30 days shelf-life after 30 days unopened storage; * *p* < 0.05, T.amb versus 5 °C storage temperature.

**Figure 5 pharmaceutics-12-00378-f005:**
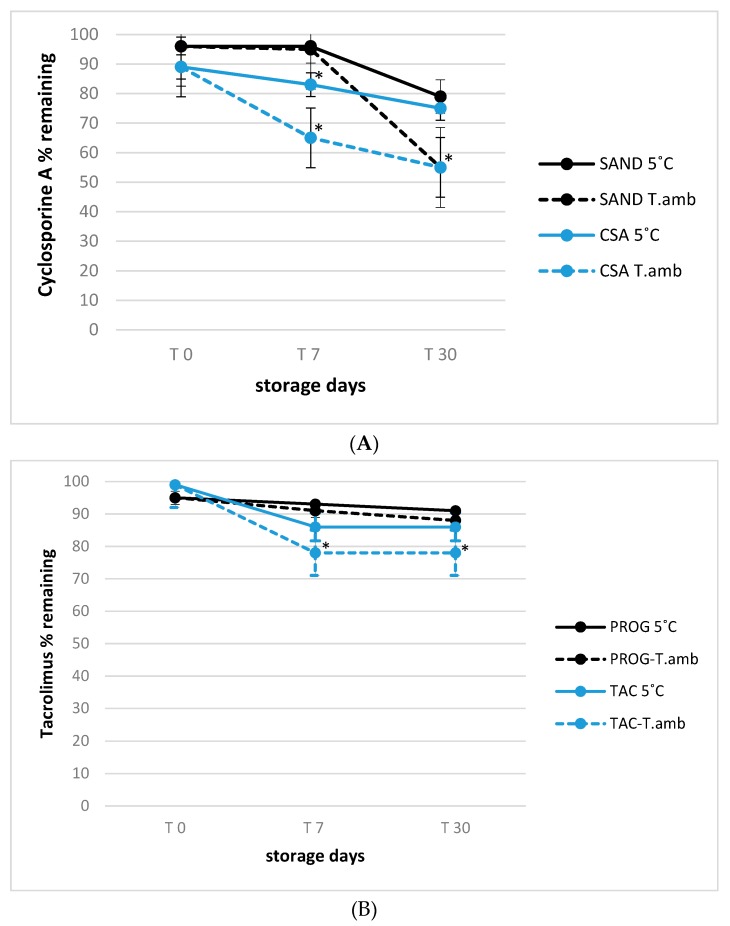
(**A**) 1.0% cyclosporine A formulations stability for 30 days shelf-life after 60 days unopened storage; * *p* < 0.05, T.amb versus 5 °C storage temperature. (**B**) 0.1% tacrolimus formulations stability for 30 days shelf-life after 60 days unopened storage; * *p* < 0.05, T.amb versus 5 °C storage temperature.

**Figure 6 pharmaceutics-12-00378-f006:**
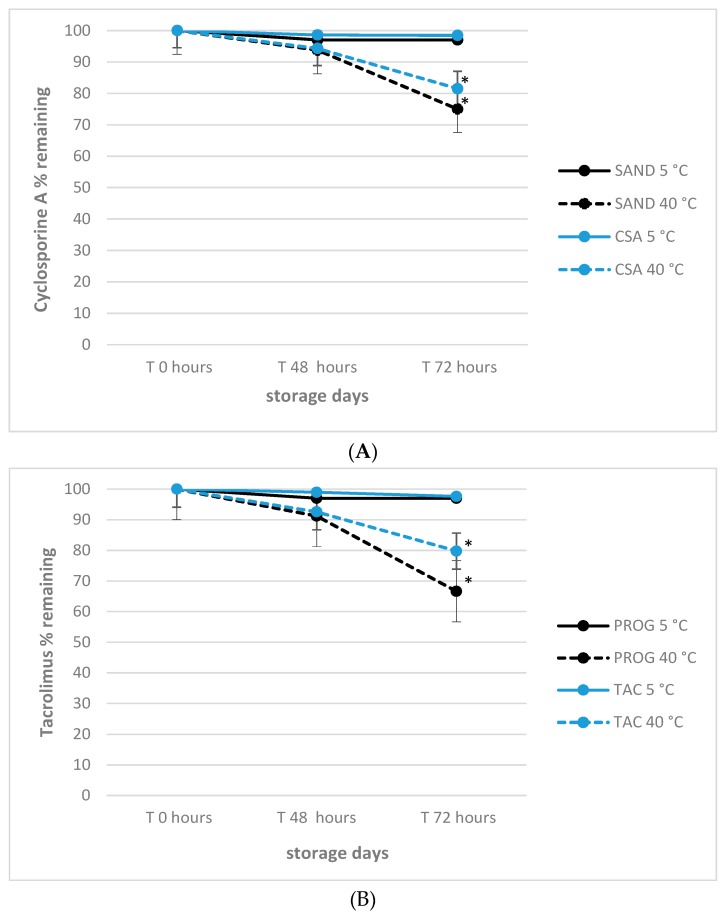
(**A**) Effect of delivery simulated conditions on 1.0% cyclosporine A stability in commercial and experimental eye drops formulations; * *p* < 0.05, T.amb versus 5 °C storage temperature. (**B**) Effect of delivery simulated conditions on 0.1% tacrolimus stability in commercial and experimental eye drops formulations; * *p* < 0.05, T.amb versus 5 °C storage temperature.

**Table 1 pharmaceutics-12-00378-t001:** Composition of Cyclosporine A- and Tacrolimus-based galenic formulations.

Formulation	Composition/Quantity		Description/Function
Cyclosporine A (1%)			
Cyclosporine A ethanol free formulation (CSA)	Cyclosporine A	100 mg	Active substance
	Polyethoxylated castor oil - Cremophor RH 40	1000 mg	Emulsifier
	Polyvinylpyrrolidone (PVP)	200 mg	Polymer/Co-Emulsifier
	Ultrapure water		to reach final volume
Cyclosporine A —classical galenic (SAND)	Sandimmun^®^ solution	50 mg/mL	Active substance— injectable solution
	Cyclosporine A	100 mg	
	Ethanol	556 mg	Solvent
	Polyethoxylated castor oil - Cremophor	1444 mg	Emulsifier
	Lacrimart^®^	8 mL:	Artificial tear
	Benzalkonium chloride		Preservative
	Methyl cellulose		Thickener
	NaCl e KCl		Osmotic agent
	Edetic acid		Chelating agent
	Ultrapure water		to reach final volume
Tacrolimus (0.1%)			
Tacrolimus ethanol free formulation (TAC)	Tacrolimus	10 mg	Active substance
	Polyethoxylated castor oil—Cremophor RH 40	1000 mg	Emulsifier
	Polyvinylpyrrolidone (PVP)	200 mg	Polymer/Co-Emulsifier
	Ultrapure water		to reach final volume
Tacrolimus— classical galenic (PROG)	Prograf^®^ solution	5 mg/mL	Active substance— injectable solution
	Tacrolimus	10 mg	
	Ethanol	1620 mg	Solvent
	Olyethoxylated castor oil—Cremophor	380 mg	Emulsifier
	Lacrimart^®^	8 mL:	Artificial tear
	Benzalkonium chloride		Preservative
	Methyl cellulose		Thickener
	NaCl e KCl		Osmotic agent
	Edetic acid		Chelating agent
	Ultrapure water		to reach final volume

**Table 2 pharmaceutics-12-00378-t002:** Compound formula, parent exact mass, main fragment mass and polarity of the two compounds under study.

Compound	Formula	Parent Exact Mass (m/z)	Main Fragment Mass (m/z)	Ionization Polarity
Tacrolimus	C_44_H_69_NO_12_	821.51580 *	768.46922	(+)
Cyclosporine A	C_62_H_111_N_11_O_12_	1219.87519 *	1202.85098	(+)
Proadifen (SFK-525A)	C_23_H_31_NO_2_	354.24276	209.13289	(+)

* The parent exact masses were calculated as ammonium adducts [M+NH_4_]^+^.

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
