# Peer review of "Stability and Safety Traits of Novel Cyclosporine A and Tacrolimus Ophthalmic Galenic Formulations Involved in Vernal Keratoconjunctivitis Treatment by a High-Resolution Mass Spectrometry Approach"

_pharmaceutics, 2020, doi:10.3390/pharmaceutics12040378_

Round 1

Reviewer 1 Report

The manuscript entitled “Stability and safety traits of novel cyclosporine A and tacrolimus ophtalmic galenic formulations involved in Vernal keratoconjunctivitis treatment” is in general interesting well written and innovative. According to my best knowledge no similar data have been published before.  However I have several comments which could be included:

Line 96

I suggest to discuss more in detail the MS spectra. Perhaps proposing a fragmentation path would be interesting.

Figure 2A, Figure 3A

Please improve the graphics (Y and X axis scales).

In general there is a lack of statistical analysis (T-test statistics, p-values etc.)

Author Response

RESPONSE TO REVIEWER 1

We gratefully acknowledge the interest in our work and the valuable comments, suggestions and corrections.

Our responses immediately follow, with indication of the changes made in the revised version of the manuscript. Moreover, in the text the corrections are highlighted in red. Moreover, all the paper has been proofread by a native speaker of English.

Author's Reply to the Review Report (Reviewer 1)

The manuscript entitled “Stability and safety traits of novel cyclosporine A and tacrolimus ophthalmic galenic formulations involved in Vernal keratoconjunctivitis treatment” is in general interesting well written and innovative. According to my best knowledge no similar data have been published before.  However I have several comments which could be included:

  • Line 96 I suggest to discuss more in detail the MS spectra. Perhaps proposing a fragmentation path would be interesting.

The MS spectra details obtained by using High Resolution Mass Spectrometry approach are insert in the manuscript to better elucidate the compounds fragmentations as well.

  • Figure 2A, Figure 3° Please improve the graphics (Y and X axis scales).

All figures are improved to make the comprehension more clear.

  • In general there is a lack of statistical analysis (T-test statistics, p-values etc.)

Statistical data treatment are insert into the manuscript to confirm proper differences among storage time points, temperature and formulation typologies as well. Stability for all investigated formulation was defined as the time at which 90% of the initial concentration of Tacrolimus and Cyclosporine A active compounds remained (t90); the initial concentration (time point 0) was considered to be 100%. Active compounds concentration was expressed as the percentage of the initial tacrolimus concentration remaining at each sampling time. Statistical evaluation of the data among sampling time and temperature was conducted by using SPSS software, Version 24. Chicago, IL: SPSS Inc; 2002. A probability value of p < .05 was considered statistically significant.

Reviewer 2 Report

This study aimed to investigate the stability and safety of the novel cyclosporine A and tacrolimus ophthalmic galenic formulations. The findings indicated that even preparing cyclosporine A powder in galenic, ethanol-free, eye drop and tacrolimus in ethanol-free, galenic eyedrop, the novel formulations were still stable, sterile and potentially effective like traditional formulations. I agree that this research issue may be important for clinicians. However, from my own experience, the relevant findings might not support the conclusion in this study. The methodology was not clear and the findings were insufficient. Therefore, I recommend authors to address clear methodology in the manuscript and provide more detail results to support their hypothesis. Followings were some major and minor concerns.

Major concerns:

  1. This study performed a serial of stability tests to demonstrate the stability of the novel formulations. However, authors provided no information regarding the Relative Humidity (RH) throughout all manuscript. Humidity is a necessary factor in the stability tests.

  1. According to the standard of ICH Q1A (R2) (Stability Testing Guidelines:Stability Testing of New Drug Substances and Products) produced by the European Medicines Agency, a complete stability study should evaluate the container stability. However, there was no relevant statements about this issue in the manuscript.

  1. This study has addressed some statements, including ‘significant decline in its concentration’ or ‘observing no significant degradative effect’. However, this study only provided the descriptive data and did not carry out any statistical analysis. Therefore, it was unclear whether there were differences between different formulations or different time points. In order to support the conclusion in this study, I recommend authors to conduct relevant statistical analyses.

Minor concerns:

  1. This study indicated that the relevant findings demonstrate the safety and stability of the formulations. However, about the safety issue, the authors only performed the microbiological stability. The relevant statements may contribute to the misleading. The readers may expect to see the risk of adverse effect about the new formulations. Therefore, the word ‘safety’ should be revised as ‘microbiological safety’ throughout all manuscript.

  1. There is a typo on the title, page 1 and page 2. ‘Ophtalmic’ should be’ ophthalmic’

Author Response

RESPONSE TO REVIEWER 2

We gratefully acknowledge the interest in our work and the valuable comments, suggestions and corrections.

Our responses immediately follow, with indication of the changes made in the revised version of the manuscript. Moreover, in the text the corrections are highlighted in red. Moreover, all the paper has been proofread by a native speaker of English.

Author's Reply to the Review Report (Reviewer 2)

This study aimed to investigate the stability and safety of the novel cyclosporine A and tacrolimus ophthalmic galenic formulations. The findings indicated that even preparing cyclosporine A powder in galenic, ethanol-free, eye drop and tacrolimus in ethanol-free, galenic eyedrop, the novel formulations were still stable, sterile and potentially effective like traditional formulations. I agree that this research issue may be important for clinicians. However, from my own experience, the relevant findings might not support the conclusion in this study. The methodology was not clear and the findings were insufficient. Therefore, I recommend authors to address clear methodology in the manuscript and provide more detail results to support their hypothesis. Followings were some major and minor concerns.

Major concerns:

  • This study performed a serial of stability tests to demonstrate the stability of the novel formulations. However, authors provided no information regarding the Relative Humidity (RH) throughout all manuscript. Humidity is a necessary factor in the stability tests.

We agree with the suggestion and insert details concerning storage conditions in material and method section in which a constant monitoring of temperature and RH value was registered.

  • According to the standard of ICH Q1A (R2) (Stability Testing Guidelines:Stability Testing of New Drug Substances and Products) produced by the European Medicines Agency, a complete stability study should evaluate the container stability. However, there was no relevant statements about this issue in the manuscript.

Containers commonly used for ophthalmic products include glass containers, and polyethylene containers. Glass containers and polyethylene (PE) containers are recognised to be superior in maintaining stability of ophthalmic preparations. LDPE container were used to store all involved formulations in the present research. The container quality represent a crucial point since the purity of a medicinal preparation may also change during storage due to leaching of chemical or chemicals into the drug preparation from the container materials, from the labels on the containers, or from the environment where the packaged is stored. Thus, containers used for packaging medicinal preparations can significantly affect the stability and purity of the preparations as well. All LPDE containers used were certified for storage of ophthalmic drug regarding criteria of water loss, environmental stress cracking resistance (ESCR). The plastic material are certified according European Pharmacopoeia (“Polyolefines and Polyethylene without additives for containers for preparations for parenteral use and for ophthalmic preparations” (8th edition – 2014)).

They were tested by the producer toward accelerated condition of temperature and relative humidity (40°C± 2°C/ NMT 25 % R H for plastic) during three months estimation drug assay. Plastic materials are tested according the migration test to assess the safety conditions toward compounds that can be transfer form plastic material to eyedrop. The stability studies were finally conducted on the drug substance packaged in a container closure system that is the same of those proposed for storage and distribution. 

  • This study has addressed some statements, including ‘significant decline in its concentration’ or ‘observing no significant degradative effect’. However, this study only provided the descriptive data and did not carry out any statistical analysis. Therefore, it was unclear whether there were differences between different formulations or different time points. In order to support the conclusion in this study, I recommend authors to conduct relevant statistical analyses.

We strongly agree with this indication. Statistical data treatment are insert into the manuscript to confirm proper differences among storage time points, temperature and formulation typologies as well. Stability for all investigated formulation was defined as the time at which 90% of the initial concentration of Tacrolimus and Cyclosporine A active compounds remained (t90); the initial concentration (time point 0) was considered to be 100%. Active compounds concentration was expressed as the percentage of the initial tacrolimus concentration remaining at each sampling time. Statistical evaluation of the data among sampling time and temperature was conducted by using SPSS software, Version 24. Chicago, IL: SPSS Inc; 2002. A probability value of p < .05 was considered statistically significant.

Minor concerns:

  • This study indicated that the relevant findings demonstrate the safety and stability of the formulations. However, about the safety issue, the authors only performed the microbiological stability. The relevant statements may contribute to the misleading. The readers may expect to see the risk of adverse effect about the new formulations. Therefore, the word ‘safety’ should be revised as ‘microbiological safety’ throughout all manuscript.

We are agree, so microbiological safety term was used throughout all manuscript.

  • There is a typo on the title, page 1 and page 2. ‘Ophtalmic’ should be’ ophthalmic’

We have corrected the title for typo.

Reviewer 3 Report

Authors have conducted an extensive LC-MS centric stability study of topical ophthalmic formulations of cyclosporin and tacrolimus. The methodology developed could be useful for general studies of these drugs in formulations. The study lacks a couple of experimental design aspects which are important:

1) the effect of temperature is not studied at 40degC; the 3rd temp condition is important from the perspective of several pharmaceutical considerations (i.e. accurate calculation of reaction rates)

2) physical stability of the formulations is not assessed, such as possibility of precipitation; follow up on "Solubilization of Cyclosporine in Topical Ophthalmic Formulations: Preformulation Risk Assessment on a New Solid Form" https://doi.org/10.1016/j.xphs.2019.06.008 Journal of Pharmaceutical Sciences
Volume 108, Issue 10, October 2019, Pages 3233-3239

3) minor typo on page 4 of 15 line 122 (*3 times per day)

Author Response

RESPONSE TO REVIEWER 3

We gratefully acknowledge the interest in our work and the valuable comments, suggestions and corrections.

Our responses immediately follow, with indication of the changes made in the revised version of the manuscript. Moreover, in the text the corrections are highlighted in red. Moreover, all the paper has been proofread by a native speaker of English.

Author's Reply to the Review Report (Reviewer 3)

Authors have conducted an extensive LC-MS centric stability study of topical ophthalmic formulations of cyclosporin and tacrolimus. The methodology developed could be useful for general studies of these drugs in formulations. The study lacks a couple of experimental design aspects, which are important:

  • the effect of temperature is not studied at 40 deg C; the 3rd temp condition is important from the perspective of several pharmaceutical considerations (i.e. accurate calculation of reaction rates)

We thank the reviewer for this comment. The accelerated condition set at 40°C was not included in the current study design since the testing was planned to evaluate the stability of tacrolimus and cyclosporine formulations in the short use term from their preparation to the end established by the one month-lasting treatment.

This choice was found comforted by the experimental observation demonstrating that already after 72 h of storage 40°C, simulating the worst transportation conditions as described in the manuscript, this storage condition is absolutely unsuitable for these drugs formulations storage.

  • physical stability of the formulations is not assessed, such as possibility of precipitation; follow up on "Solubilization of Cyclosporine in Topical Ophthalmic Formulations: Preformulation Risk Assessment on a New Solid Form" https://doi.org/10.1016/j.xphs.2019.06.008 Journal of Pharmaceutical Sciences Volume 108, Issue 10, October 2019, Pages 3233-3239

We agree with the suggestion also according to the above reported research in which polymeric nanomicelles provide a promising strategy for ocular delivery of CSA because of their potential in enhancing bioavailability, improving tolerability, and reducing systemic side effects.

As general consideration, the use of mixtures of surfactants and polymers would prevent the conversion of the metastable forms to the thermodynamically stable form. In addition, caution should be taken to guarantee the drug stability over the shelf life of the drug product and to withstand temperature excursions. Based on above mentioned considerations, in order to confirm the stability of all formulations involved in the present research (Tacrolimus and CSA based), they are undergo to LC-HRMS evaluation to detect the concentration of active compounds at time 0. For each formulation, concentration reaching more than 99.5% of active compound compared with the declared one was considered as satisfactory criteria. In its final form, the CSA formulation consists of an isotonic aqueous solution with micelles of Cremophor® solubilizing the cyclosporine with surfactant and polymer aiding solubility. During all experiments all multidose eyedroppers were emptied into polycarbonate test tubes and the solutions were visually inspected under white light in front of a matt black panel and a non-glare white panel. Aspect and colour of the solutions were noted, and a screening for visible particles, haziness, or gas development was performed. All considerations are reported into the manuscript

  • minor typo on page 4 of 15 line 122 (*3 times per day)

We have correct the typo.

Reviewer 4 Report

General Comments to the Authors

The authors study compounded ophthalmic formulations of cyclosporin A and tacrolimus. The author’s goal is to identify an ethanol-free formulation. The authors conduct analytical method development LC-MS, prepare formulations and conduct stability evaluation under shelf-life and in-use conditions.

Specific Comments to the Authors

  1. The authors report much analytical method development but this is not reflected in the title.
  2. Section 4.3.b: Be more specific as to the composition of the formulation. What do you mean by “…with an excipient suitable for….”? Suggest a table of compositions.
  3. The Figures: Suggest better color and symbol coding, and to have each pair the same color with different symbols for the temperatures.
  4. The Figures: y-axis needs rewording
    1. 2A and 5B: change “Tacrolimus degradation percentage (%)” to “Tacrolimus % remaining”
    2. 2B and 5A: change “Cyclosporin A degradation percentage (%)” to “Cyclosporin A % remaining”
    3. 3: change “Reduction percentage (%)” to “% Degradation”
    4. 4A: and 6A change “Cyclosporin A reduction percentage (%)” to “Cyclosporin A % remaining”
    5. 4B and 6B: change “Tacrolimus reduction percentage (%)” to “Tacrolimus % remaining”
  5. Is Table 2 needed as all entries are “sterile”?

Author Response

RESPONSE TO REVIEWER 4

We gratefully acknowledge the interest in our work and the valuable comments, suggestions and corrections. Our responses immediately follow, with indication of the changes made in the revised version of the manuscript. Moreover, in the text the corrections are highlighted in red. Moreover, all the paper has been proofread by a native speaker of English.

Author's Reply to the Review Report (Reviewer 4)

The authors study compounded ophthalmic formulations of cyclosporin A and tacrolimus. The author’s goal is to identify an ethanol-free formulation. The authors conduct analytical method development LC-MS, prepare formulations and conduct stability evaluation under shelf-life and in-use conditions.

Specific Comments to the Authors

  • The authors report much analytical method development but this is not reflected in the title.

We have modify the title to better focalise the research in which the HRMS analytical method was used to investigate active compounds in the new and ethanol-free proposed formulations.

  • Section 4.3.b: Be more specific as to the composition of the formulation. What do you mean by “…with an excipient suitable for….”? Suggest a table of compositions.

We have create a new table for all investigated formulations to present the composition clear as possible.

  • The Figures: Suggest better color and symbol coding, and to have each pair the same color with different symbols for the temperatures.

All figures are modified according to suggestions to make the results clear as possible. Each pair of formulations had the same colour with different line trait (continuous and trace) in relation to their storage temperature (continuous for 5 degree and trace for room temperature).

  • The Figures: y-axis needs rewording.

All figures y-axis were reworded.

               2A and 5B: change “Tacrolimus degradation percentage (%)” to “Tacrolimus % remaining”

               2B and 5A: change “Cyclosporin A degradation percentage (%)” to “Cyclosporin A % remaining”

               3: change “Reduction percentage (%)” to “% Degradation”

               4A: and 6A change “Cyclosporin A reduction percentage (%)” to “Cyclosporin A % remaining”

               4B and 6B: change “Tacrolimus reduction percentage (%)” to “Tacrolimus % remaining”

               All figures were completely redone inserting the new suggestions into the legends as well.

  • Is Table 2 needed as all entries are “sterile”?

The table was deleted describing the results in a paragraph.

Round 2

Reviewer 2 Report

The authors have made a number of changes to their manuscript in response to the reviewer’s comments. The quality of English throughout the manuscript has greatly improved and the authors also provided clear statements about the temperature, RH value, and container stability. The authors further revised the unsuitable statements throughout all manuscript. However, I still have a few queries that should be addressed before recommending for publication. 

--Authors have addressed that they conducted relevant statistical analyses in this study. Statistical evaluation of the data among sampling time and temperature was conducted by using SPSS software. However, the authors did not state which types of statistical analyses were used in this study. The authors should provide relevant information. Additionally, I’m not sure which types of statistical analyses were used in this study. However, doing multiple times of T test is not recommended. If author attempted to investigate the differences between different formulations or different time points, repeated ANOVA may be much more suitable. I recommend authors to ask the biostatisticians for help and provide clear information for the readers.

--The authors have provided the information about the p value in the manuscript. However, the relevant statements were not precise. For example, “The results confirm the negative impact of high temperatures also during delivery, with significant decay (p<0.01) observed after 48 h for products stored at 40°C”. It was still unclear which groups had significant decay or which time point showed the significant decay. Authors should check and revise all relevant statements once again.

--Additionally, I recommend authors to provide the symbols about the statistical significance on the figures.

Author Response

RESPONSE TO REVIEWER 2

We gratefully acknowledge the interest in our work and the valuable comments, suggestions and corrections.

Our responses immediately follow, with indication of the changes made in the revised version of the manuscript. Moreover, in the text the corrections are highlighted in red. Moreover, all the paper has been proofread by a native speaker of English.

Author's Reply to the Review Report (Reviewer 2)

The authors have made a number of changes to their manuscript in response to the reviewer’s comments. The quality of English throughout the manuscript has greatly improved and the authors also provided clear statements about the temperature, RH value, and container stability.

The authors further revised the unsuitable statements throughout all manuscript. However, I still have a few queries that should be addressed before recommending for publication. 

--Authors have addressed that they conducted relevant statistical analyses in this study. Statistical evaluation of the data among sampling time and temperature was conducted by using SPSS software. However, the authors did not state which types of statistical analyses were used in this study. The authors should provide relevant information. Additionally, I’m not sure which types of statistical analyses were used in this study. However, doing multiple times of T test is not recommended. If author attempted to investigate the differences between different formulations or different time points, repeated ANOVA may be much more suitable. I recommend authors to ask the biostatisticians for help and provide clear information for the readers.

We have better detailed the data treatment. In particular, the results were processed by one-way analysis of variance (ANOVA); Analysis of variance (ANOVA) was applied to the data and significant differences were determined by 1-way ANOVA and Student-Newman-Keuls (SNK) as post-hoc test to evaluate differences among formulations and among storage times. An effect was considered significant at the 5% level (< 0.05).

-The authors have provided the information about the p value in the manuscript. However, the relevant statements were not precise. For example, “The results confirm the negative impact of high temperatures also during delivery, with significant decay (p<0.01) observed after 48 h for products stored at 40°C”. It was still unclear which groups had significant decay or which time point showed the significant decay. Authors should check and revise all relevant statements once again.

We have better explained concerning the proper sentences and statements in the manuscript, the significances obtained by data comparison to present the results more clear. In detail, we have better formulated the sentences regarding influence of storage times on active compounds decay. The differences observed among formulations are associated to (*) as symbol when significant into figures.

-Additionally, I recommend authors to provide the symbols about the statistical significance on the figures.

According the the suggestion, symbols about significances are provided in the figures.

Reviewer 3 Report

no comments

Author Response

RESPONSE TO REVIEWER 3

We gratefully acknowledge the interest in our work and the valuable comments, suggestions and corrections.  Our responses immediately follow, with indication of the changes made in the revised version of the manuscript. Moreover, in the text the corrections are highlighted in red. Moreover, all the paper has been proofread by a native speaker of English.

Author's Reply to the Review Report (Reviewer 3)

Open Review

English language and style

( ) Extensive editing of English language and style required 
( ) Moderate English changes required 
(x) English language and style are fine/minor spell check required 
( ) I don't feel qualified to judge about the English language and style 

Comments and Suggestions for Authors

no comments

Considering no comments, we confirm the paper was revised by a native English speaker

Round 3

Reviewer 2 Report

The authors have answered most questions that were raised by me. My recommendation now would be to accept.